# Computer simulations of food oral processing to engineer teeth cleaning

C.G. Skamniotis[1], M. Elliott[2] & M.N. Charalambides[1]

Oral biofilm accumulation in pets is a growing concern. It is desirable to address this problem via non-invasive teeth cleaning techniques, such as through friction between teeth and food during chewing. Therefore, pet food design tools are needed towards optimising cleaning efficacy. Developing such tools is challenging, as several parameters affecting teeth cleaning should be considered: the food's complex mechanical response, the contacting surfaces topology as well as the wide range of masticatory and anatomical characteristics amongst breeds. We show that Finite Element (FE) models can efficiently account for all these parameters, through the simulation of food deformation and fracture during the first bite. This reduces the need for time consuming and costly in-vivo or in-vitro trials. Our in-silico model is validated through in-vitro tests, demonstrating that the initial oral processing stage can be engineered through computers with high fidelity.

[1] Department of Mechanical Engineering, Imperial College London, London SW7 2AZ, UK. [2] Mars Petcare, Oakwell Way, Birstall, Batley WF17 9LU, UK. Correspondence and requests for materials should be addressed to C.G.S. (email: christos.skamniotis@eng.ox.ac.uk) or to M.N.C. (email: m.charalambides@imperial.ac.uk)

By the age of three, 80% of dogs and 70% of cats develop oral biofilms e.g., plaque and tartar[1]. This occurs typically through the adherence of food particles, bacteria and saliva onto the teeth, especially in regions surrounding the gum line[2–4]. Bacteria can be absorbed into the bloodstream via bleeding gums, causing organ infections[5], while tartar tends to push the gum away from the teeth roots, eventually leading to teeth loss[6,7]. Amongst various measures employed to date e.g., dental prophylaxis and oral hygiene gels[4,8], tooth brushing is inarguably the most effective. Nevertheless, the latter requires significant effort by the pet owners[3,9,10]. As a result, the idea of biofilm removal via natural teeth-food mechanical abrasion during oral processing gains significant popularity in oral pet care[8,9].

However, no systematic measurement method for cleaning efficacy exists, while designing the texture-size-shape of oral care foods to serve a range of breeds appears to be a complex multi-parametric problem[10]. In particular, the deformation and fracture patterns in the food dictate whether teeth-food interaction will reach teeth areas surrounding the gum line. Simultaneously, the food rigidity/stiffness designates firstly the chewing force magnitudes involved[11], thus the severity of mechanical teeth-food interaction, while the food fracture toughness determines the number of chews required for food comminution and subsequent swallowing[12] i.e., the amount of teeth-food interaction per unit calorie consumption. Lastly, the diverse dental anatomy amongst breeds[13] implies variation in ways of chewing[10,14] and thus also food textural preferences[15–17].

Although in-vivo teeth cleaning trials are reported[4,8,9,18], the level of understanding of the actual biofilm removal mechanism remains limited, while the role of food texture has not been studied. Concurrently, in-vitro investigations[19–21] impose challenges in physically reproducing physiological conditions and disparate anatomies[20–22]. In contrast, recent reports[10,11] suggest that an in-silico approach can facilitate straightforward studies of food texture-size-shape to optimise cleaning efficacy for individual breeds; this would reduce the need for subjecting animals to frequent anaesthesia, as well as the need for physical food prototyping[15,23]. Specifically, virtual models based on Finite Element (FE) analysis can be developed in order to simulate how the food deforms, fractures and interacts with digital teeth representations.

A number of challenges in simulating mechanical food oral breakdown, however, remain yet to be resolved[24]; these mainly associate with remarkable mathematical complexity imposed by the material, geometric and contact non-linearities[25]. Such numerical issues were previously partially rectified through Smoothed Particle Hydrodynamics (SPH) models of mastication[25–27]. These models, however, assumed largely simplified food mechanical constitutive laws e.g., elastic-perfectly plastic stress-strain behaviour coupled with a brittle fracture criterion, thus ignoring material time dependency effects and the critical fracture toughness parameter, $G_c$[10,11,28]. On the other hand, although realistic constitutive laws for foods have been established[29–32], to date these are mostly implemented in modelling a linear, predefined fracture path, which cannot be assumed in food oral breakdown[10,11].

This study presents an oral processing FE model, able to capture both the actual viscoplastic-damage constitutive behaviour, as well as the non-linear fracture path that occur in a starchy food during the first bite. The numerical predictions are validated through experimental data obtained from the physical replicate of the mastication model, while the viscoplastic-damage constitutive law is constructed based on extensive mechanical characterisation of the starchy food. We utilise the FE model to extract key information on the effect of food geometry on teeth cleaning efficacy. The model provides an important step forward

towards gaining control over the biofilm removal process whilst taking into account the masticatory and nutritional needs of each pet breed.

## Results

**Food damage-fracture processes.** The food extrudates consist of (w/w): 47% cereal starch, 45.5% water, 2.5% cellulose fibres (500 μm maximum length) and 5% minerals. Figure 1a shows the four-point bending test set up inside a Scanning Electron Microscopy (SEM) chamber. Uniaxial compression tests (see Fig. 1d) are performed in a similar manner. Figure 1b reveals that damage in bending involves micro-cracking in the volume of the specimen under tensile deformation (specimen upper region). The micro-cracks are perpendicular to the direction of tensile strain, in agreement with Fig. 1e which shows that in compression, excessive micro-cracking occurs along the direction of compressive loading which is normal to the lateral tensile strain; the latter is induced by the significant specimen expansion associated with very low compressibility i.e., Poisson's ratio, $v = 0.5$, as it was previously determined in a similar starch-fibre recipe[10]. The above agreement supports the use of tensile strain (i.e., maximum principal strain) in the damage onset criterion of the food constitutive law. Simultaneously, Fig. 1c illustrates that macroscopic fracture begins when the micro-cracks shown in Fig. 1b grow and coalesce into a macro-crack. The latter propagates with significant fibre-bridging involved (see Fig. 1c), a mechanism which has been reported[33] to enhance fracture toughness, $G_c$ (energy dissipation per unit crack surface growth[28]) in this material; the latter added to the tough nature of starch, clearly suggests that $G_c$ is a critical parameter in describing damage/fracture in this food.

**Food viscoplastic-damage constitutive behaviour.** Figure 1 panels f–h and i–l depict specimen behaviour during tensile and compression tests, respectively, which are performed to obtain equivalent stress-strain, $\sigma_{eq} - \varepsilon_{eq}$, data. The data are summarised in Fig. 2, together with the viscoplastic-damage constitutive law fits: the monotonic data for six constant equivalent strain rates, $\dot{\varepsilon}_{eq} = 0.0001\,\text{s}^{-1}, 0.001\,\text{s}^{-1}, 0.01\,\text{s}^{-1}, 0.1\,\text{s}^{-1}, 1\,\text{s}^{-1}, 5\,\text{s}^{-1}$, are plotted in Fig. 2a, d), for compression and tension, respectively; the relaxation data at constant strains of $\varepsilon_{eq} = 0.1, 0.2, 0.5$ in compression are reported in Fig. 2b, while the corresponding data at $\varepsilon_{eq} = 0.03, 0.06, 0.08$ in tension are shown in Fig. 2e; loading-unloading data at two rates, $\dot{\varepsilon}_{eq} = 0.1\,\text{s}^{-1}, 1\,\text{s}^{-1}$ and a maximum $\varepsilon_{eq} = 0.2, 0.5, 0.8$ in compression are plotted in Fig. 2c, while the corresponding data at the same rates and a maximum $\varepsilon_{eq} = 0.07, 0.2, 0.22$ in tension are illustrated in Fig. 2f. The monotonic compression data (Fig. 2a) are plotted up to the onset of specimen barrelling and/or micro-cracking effects (as shown in Fig. 1e, k–l), the latter being a function of $\dot{\varepsilon}_{eq}$. Four main attributes are found: (i) strongly non-linear and time-dependent $\sigma_{eq} - \varepsilon_{eq}$ behaviour, (ii) higher stresses in tension when compared to compression for common $\varepsilon_{eq}$, (iii) highly hysteretic behaviour (Fig. 2b, e) and (iv) rate dependent tensile failure strain, $\varepsilon_f(\dot{\varepsilon}_{eq})$ (last data points denoted as '$x$' in Fig. 2d). Significant hysteresis and non-linearity are attributed to the inherent time-dependency of starch, as well as micro-cracking which progressively degrades the material stiffness with increasing strain. Instead, the higher stiffness in tension as opposed to compression owes to a less effective fibre stiffening effect in compression[34]. Lastly, the increase of $\varepsilon_f$ with $\dot{\varepsilon}_{eq}$ relates again to micro-cracking, on the basis that even upon fixed tensile deformation micro-cracks are observed to grow with time, promoting the formation of a macro-crack i.e., ultimate failure. This is supported by the strong relationship ($R^2 = 0.97$)

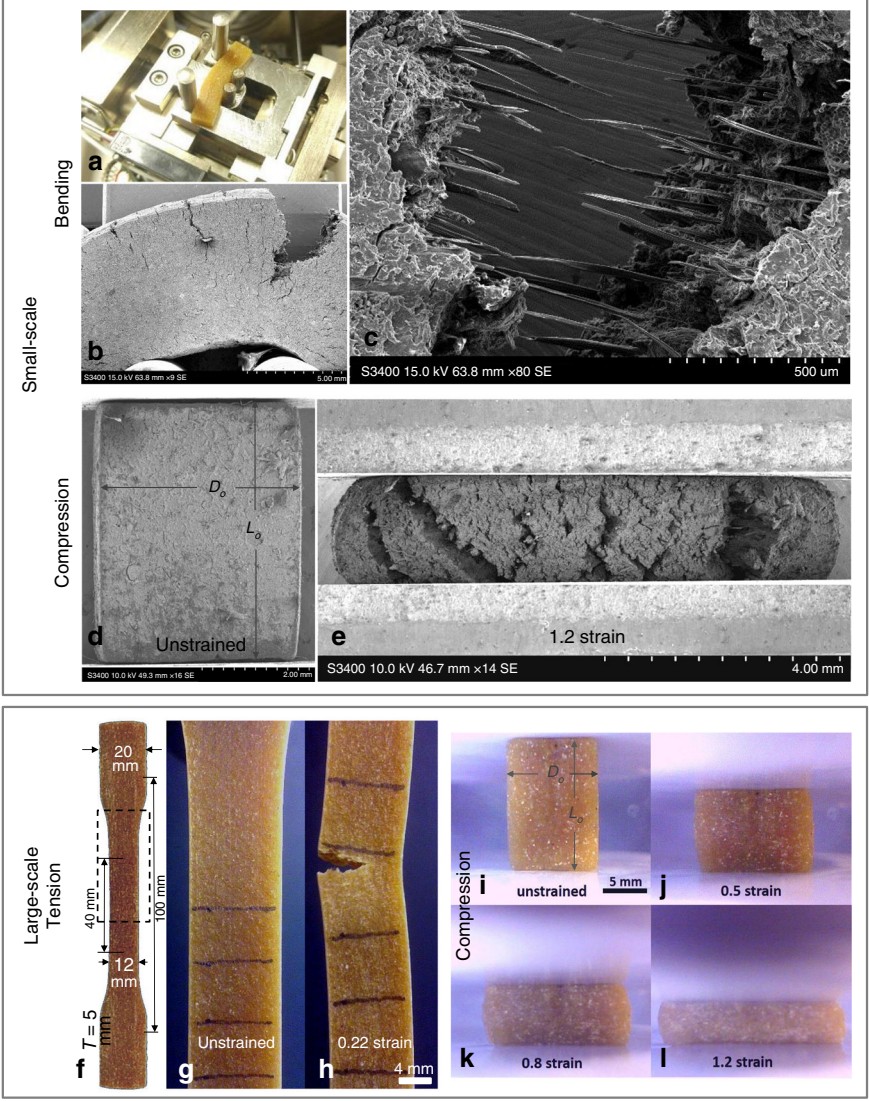

**Fig. 1** Illustration of food mechanical characterisation. Small scale tests reveals damage/fracture mechanisms, whereas large scale tests provide stress-strain data as necessary key input parameters for the material constitutive law. **a** shows in-situ four point bending test under Scanning Electron Microscopy (SEM) (vacuum chamber is open for illustration–Hitachi S-3400N SEM equipment with load cell capacity of 200 N is used) with parameters: specimen length 30 mm, width 7 mm, thickness 4 mm, span length 7 mm, striker radius 4 mm, support radii 5 mm, striker speed 2.5 mm min$^{-1}$, **b** crack propagation–tensile failure at striker displacement 3.8 mm, **c** crack faces detail highlight fibre-bridging toughening mechanism. **d** shows in-situ compression SEM test at zero strain (specimen height × square edge = 5 × 4 mm), **e** specimen axial cracking and barrelling effects; due to significant barrelling the respective stress-strain data are not valid. **f** depicts tensile specimen (effective length × width × thickness = 100 × 12 × 5 mm—dashed box denotes optical window used to track strain), **g** optical frame of specimen at zero strain, **d** optical frame at tensile failure at 0.22 strain. **i–l** show optical frames of lubricated compression on cuboid specimens (height × square edge = 13 × 8 mm). For large scale tests (panels (**f–l**)) an Instron 5543 universal testing machine with load cell capacity of 1 kN is used

revealed between the monotonic $\varepsilon_f(\dot{\varepsilon}_{eq})$ data obtained from Fig. 2d and the integration of strain history, $\varepsilon_{eq}(t) = \dot{\varepsilon}_{eq}t$, over time:

$$\varepsilon_f\left(\dot{\varepsilon}_{eq}\right) = -0.024 \ln\left(\int_0^{t_f} \dot{\varepsilon}_{eq}t\,dt\right) + 0.2 \tag{1}$$

where $t_f$ is time elapsed from $\varepsilon_{eq} = 0$ until failure ($\varepsilon_{eq} = \varepsilon_f$); for non-monotonic loading a general strain history $\varepsilon_{eq}(t)$ must be integrated in Eq. (1), instead of $\dot{\varepsilon}_{eq}t$, and this is utilised to define the tensile strain threshold value in the damage onset criterion of the viscoplastic-damage constitutive law. On the other hand, the

damage evolution component of the law is governed by the true $G_c = 0.93$ kJ m$^{-2}$ determined for this material in ref. [35], for a typical average crack speed of the order of 5 mm s$^{-1}$; the latter is of the same order of magnitude with the average crack speed of 10 mm s$^{-1}$ (estimated via FE) occurring here during food separation, for the bite speed of $\dot{\delta} = 16.6$ mm s$^{-1}$.

The viscoplastic component of the law involves an initial small elastic regime governed by elastic modulus, $E = 50$ MPa, followed by a time and stress state dependent elastoplastic regime, which thereafter degrades to capture damage in tension. This representation gives an excellent experimental-model fit in terms of the monotonic response (Fig. 2a, d) and reasonable model predictions for stress relaxation (Fig. 2c, f). Some discrepancy regarding

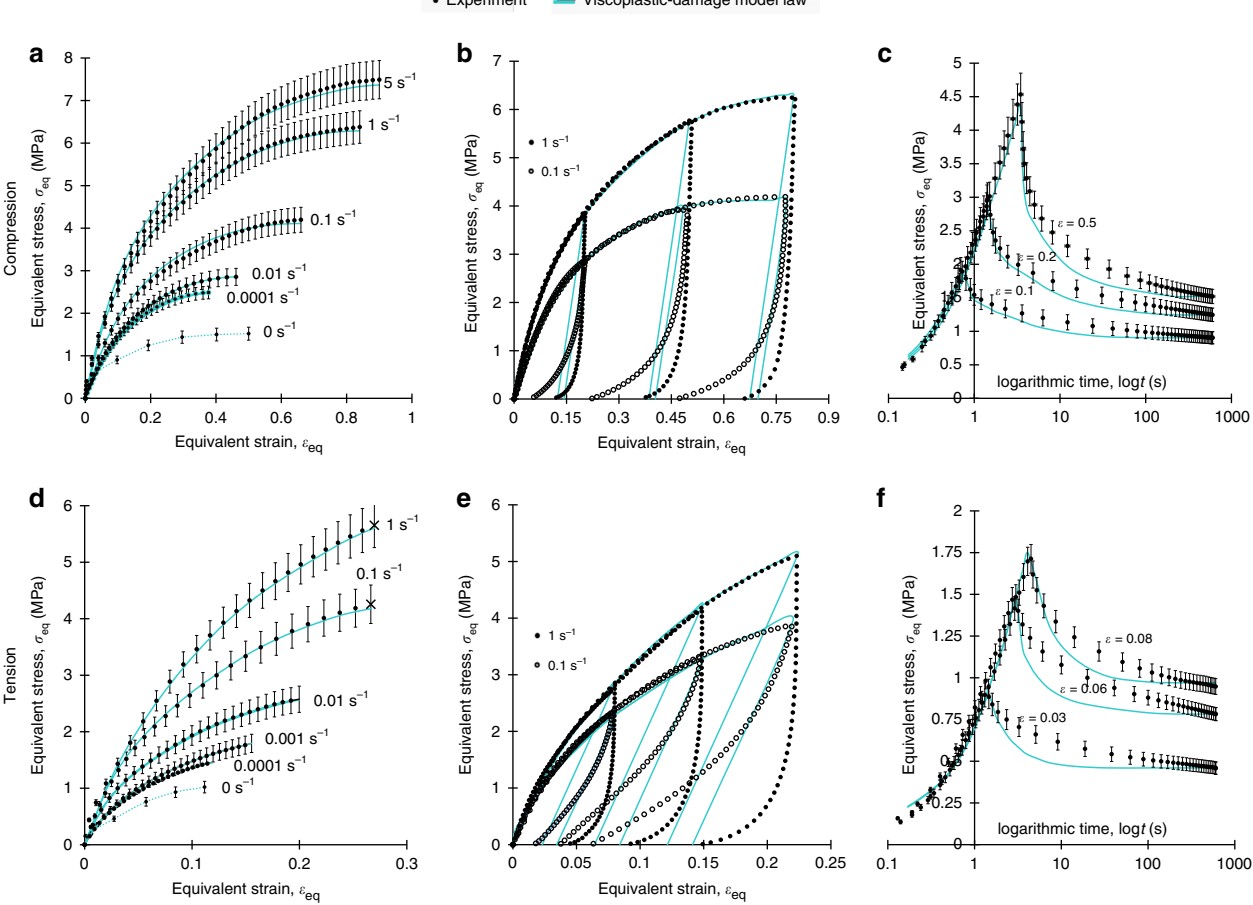

**Fig. 2** Equivalent stress versus equivalent strain data together with the viscoplastic-damage law fit. **a** depicts monotonic compression, **b** loading-unloading in compression, **c** relaxation in compression, **d** monotonic tension, **e** loading-unloading in tension, **f** relaxation in tension. Error bars denote percentage experimental error. The 'x' marker of last data points in monotonic tensile curves denotes ultimate specimen failure. The data for 0 s$^{-1}$ strain rate in **a**, **d** correspond to the last data points (long term stresses) in **c**, **f** respectively. The viscoplasticdamage law is implemented in ABAQUS[39]

the unloading response and particularly in tension (Fig. 2b, e), is attributed to the model's assumption of both time independent and stress state independent elastic unloading determined by $E = 50$ MPa, which does not account for the fact that the tensile response is generally stiffer than the compressive i.e., unloading from the same absolute maximum strain begins from higher stresses in tension than in compression. Whether tension also involves larger cyclic hysteresis due to a potentially larger extent of damage mechanisms e.g., micro-cracking, fibre-matrix debonding, is unknown.

**Comparison between physical and virtual food oral breakdown**. Figure 3 illustrates the mastication FE model construction steps. Figure 4 summarises the corresponding FE results, for a range of tooth-food friction coefficients, $\mu$, in comparison to the experimental data obtained from the physical replicate of the model (see Fig. 4b); the latter consists of the 3D-printed (stainless steel) fixtures shown in Fig. 3b, c, as well as the food specimen used to construct the virtual food item. We focus on the case of a linear occlusal vector of the mandible (lower jaw) against the maxilla (upper jaw)[36] at a displacement rate of $\dot\delta = 16.6$ mm s$^{-1}$; additional rates of $\dot\delta = 0.016$ mm s$^{-1} = 0.16$ mm s$^{-1}$ and 1.66 mm s$^{-1}$ are applied in the experiment (see Fig. 4a), and $\dot\delta = 1.66$ mm s$^{-1}$, 166.6 mm s$^{-1}$ in the FE model (presented in Supplementary Fig. 1). Figure 4a compares the force-displacement, $F \sim -\delta$, raw experimental data against the FE model predictions

(Supplementary Fig. 2 reports on FE mesh sensitivity). The strong time dependency found in the $\sigma_{eq} - \varepsilon_{eq}$ and $\varepsilon_f(\dot\varepsilon)$ food properties is reflected here in the experiment through markedly higher both maximum force, $F_{max}$, and corresponding displacement, $\delta_{F_{max}}$, required for breakdown with increasing $\dot\delta$ (Fig. 4a). The stages I−IV denoted in Fig. 4a are synchronised with the experimental-FE model frames shown in Fig. 4b (for $\dot\delta = 16.6$ mm s$^{-1}$ and $\mu = 0.3$) and correspond to: (I) onset of teeth indentation by cusps A, B (see Fig. 4b), (II) change in the $F − \delta$ slope due to inclusion of cusps C, D (see Fig. 4b) in contact with the food, (III) crack initiation i.e., onset of breakdown, at $\delta = \delta_o$, and lastly (IV) end of the test at $\delta = \delta_{max}$. The tough texture of the starchy food associates with the fact that both in the experiment and FE model, the $\delta_o$ is slightly lower than $\delta_{F_{max}}$, and that complete food separation does not occur. Instead, between stages III−IV the force drops and then increases due to one of the two food fragments being squeezed between the lower carnassial and the maxilla surface where in reality the gum would exist. This effect is not investigated for two reasons: (i) its occurrence depends on food specimen length, (ii) it is unlikely to occur in-vivo, where in fact food repositioning by the tongue would prevent hard contact between the gum and the food, towards more palatable and efficient food breakdown[25–27,37].

Figure 4a indicates a very close experimental-FE model agreement. However, this is specific to $\mu = 0.3$, since increasing $\mu$ enhances significantly $F_{max}$ and leads to a more brittle

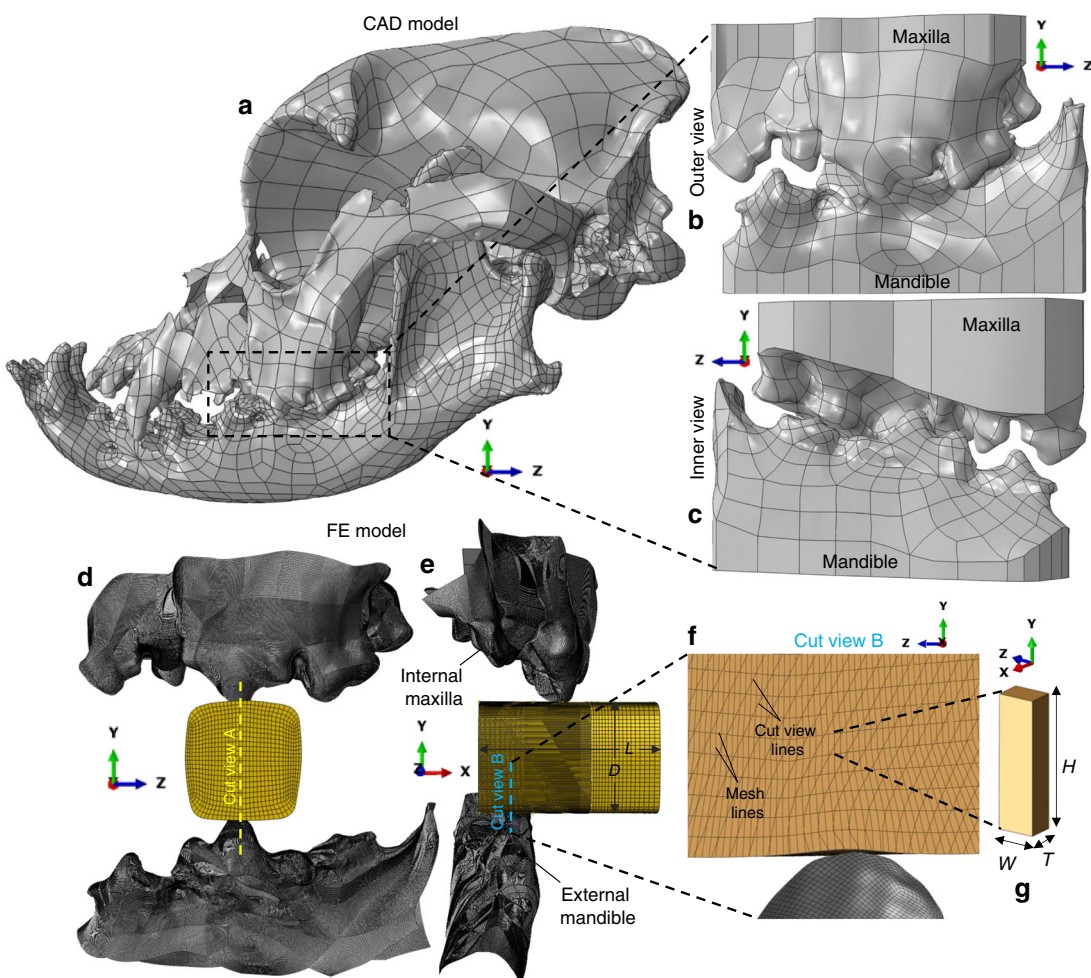

**Fig. 3** Illustration of the mastication FE model construction steps. **a** shows side view of the boxer skull CAD geometry imported in ABAQUS CAE[39]; dashed box denotes the cropped region used in the FE model. **b**, **c** show outer and inner side view of cropped skull geometry, respectively; both jaw surfaces are modified by adding flat faces in the remote of the teeth in order to obtain a closed volume essential for the 3D-printing process. **d**, **e** depict front and side views, respectively, of the FE meshed model assembly, along with positions of cut views A (used in Fig. 4d) and B and food item dimensions: length 20 mm, and effective diameter 12.6 mm. **f** and **g** correspond to cut view B showing food mesh design and element configuration details, defined by dimensions: height 0.3 mm, width 0.1 mm and thickness 0.05 mm

breakdown response i.e., sharper drop in $F$ after the onset of breakdown ($\delta > \delta_o$). This highlights that the parameter $\mu$ and potentially also the characteristics of the underlined friction law, generally influence significantly the FE predictions of the first bite response. However, characterising the details of in-vivo tooth-food interaction, as well as of the friction between the 3D-printed surfaces and the extrudates, can be challenging; this explains why using a simple friction law was instructive for the purposes of this study. The parameter $\mu$ has been measured between teeth surface substitutes i.e., Hydroxyapatite (HA) disks, and the external food extrudate surfaces, via sliding tests; these deduced $\mu = 0.2$[38]. Consequently, $\mu = 0.3$ is a likely condition to occur in the experiment, given the noticeably rougher 3D-printed surfaces compared to the HA disks. For $\mu = 0.3$ the model gives slightly lower $\delta_{F_{max}}$ than the experiment, potentially attributed to the fact that the enforced food external surface condition $G_c = 0$ kJm$^{-2}$, may underpredict the food's resistance to crack initiation. Specifically, Fig. 4c shows cracks initiating at mainly three element layers (denoted as 1, 2, 3), as soon as damage onset is satisfied ($d > 0$) at elements of the lateral food surfaces. Nevertheless, thanks to the $G_c = 0$ kJ m$^{-2}$ condition, crack propagation (through further element deletions) is

predicted correctly along a single crack, as demonstrated in Fig. 4e, which agrees with the experimental observations (Fig. 4b-IV). These results verify the model's capability and accuracy in predicting food oral breakdown at large deformations, without previous knowledge of the crack patterns, and, based on experimental $G_c$ values.

The FE model reveals that damage onset (Fig. 4b, c) is triggered by tensile strains, $\varepsilon_1$, which are found to occur at a large degree due to the low material compressibility i.e., $v = 0.475$. This simultaneously explains the fracture mechanism (at $\delta = \delta_o$) in the experiment (Fig. 4b-III). The critical strains, $\varepsilon_1$, are applied along the food length direction (denoted in Fig. 4c), which based on the currently used $L_{ch}$ definition implies $L_{ch} \approx T$. Therefore, the critical element dimension based on which damage evolution is calculated is the thickness $T = 0.05$ mm (see Fig. 3f, g). Whether the viscoplastic-damage food constitutive law is calibrated accurately is assessed through the stress triaxiality, $\eta$, equivalent strain, $\varepsilon_{eq}$, and equivalent strain rate, $\dot{\varepsilon}_{eq}$, fields. Figure 4d (corresponds to cut view A shown in Fig. 3d) reveals that $\eta$ ranges from $\eta = -2.7$ (confined compression) underneath the indentation tips, up to $\eta = 0.9$ (multiaxial tension) in the food surfaces surrounding the indentation regions. Although $\sigma_{eq}-\varepsilon_{eq}$ data were not collected for such $\eta$ values, these values

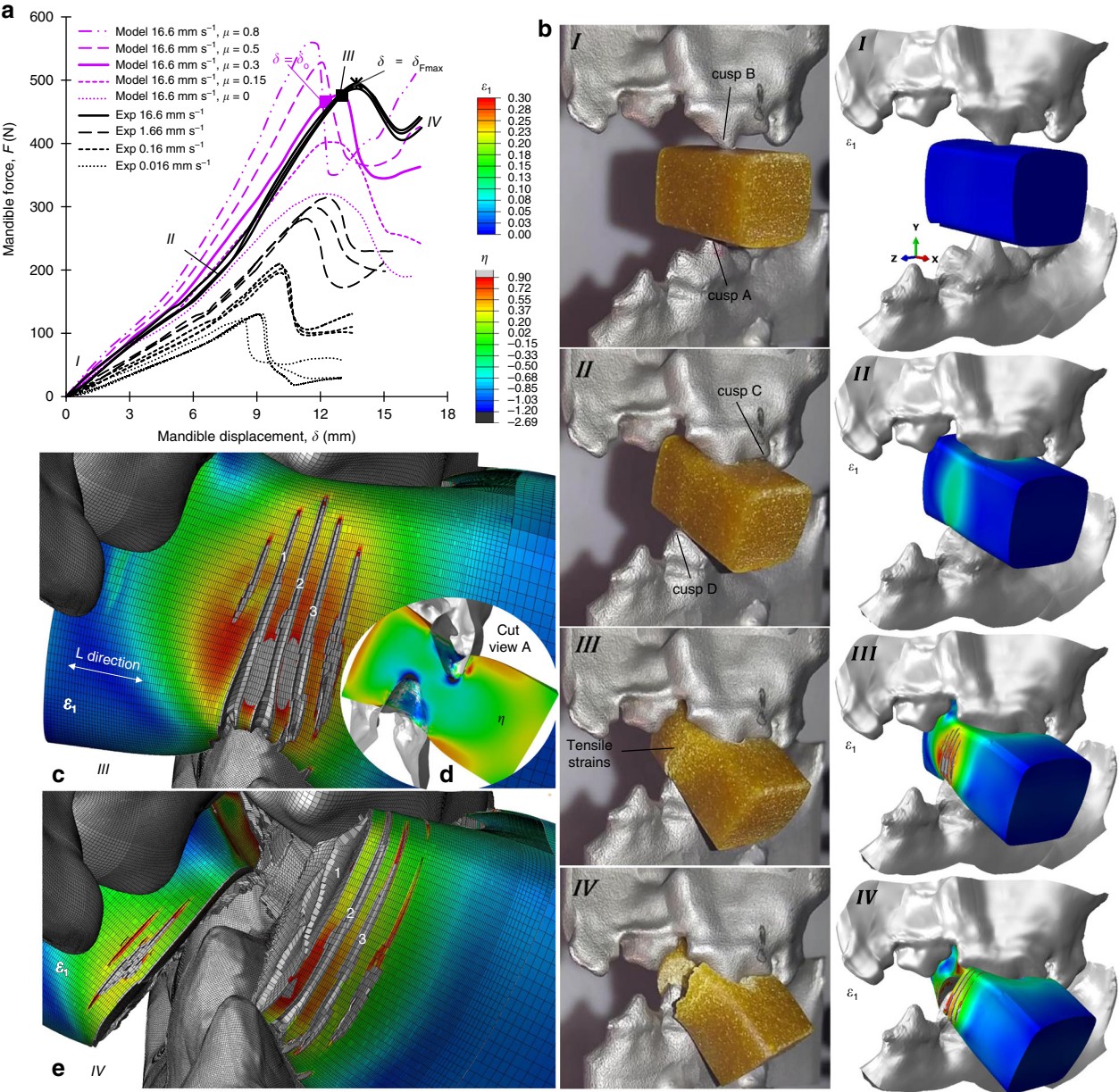

**Fig. 4** Summary of mastication experimental-FE model predictions. **a** shows raw experimental force-displacement data for the four mandible speeds together with predictions for the speed 16.6 mm s⁻¹ as a function of five friction coefficients; experiments are performed in an Instron 5543 universal testing machine with load cell capacity of 1 kN, whereas simulations are performed via ABAQUS-Explicit[39]. **b** shows qualitative experimental-model comparison for the speed 16.6 mm s⁻¹ and friction coefficient 0.3, through video frames of maximum principal strain contours corresponding to the four stages, I–IV, denoted in **a**. **c** illustrates maximum principal strain contours through side view of the food at the onset of breakdown–stage III; initial element deletions due to $G_c = 0$ kJ m⁻² mark mainly three potential crack paths, denoted as 1, 2 and 3. **d** displays stress triaxiality contours via cut view B (defined in Fig. 4d) at stage III. **e** shows progressed stage of panel **c** at maximum mandible displacement (stage IV); note that only crack 1 propagates

concern small portions of the food; instead, the dominant range is $-2/3 < \eta < 0$ which falls close to the calibrated range $-1/3 < \eta < 1/3$ of the food constitutive law. Similarly, excessive deformations i.e., $\varepsilon_{eq} = 2$, are only found to occur in small portions of the food, specifically at the regions underneath the indentation tips (where also $\eta = -2.7$ occur in Fig. 4d), as well as at the elements undergoing damage. Instead, values of $\varepsilon_{eq} < 1$ are observed in the rest of the food, such that the applied compressive strains did not exceed significantly the calibrated range $\varepsilon_{eq} < 0.9$ (Fig. 2a for s⁻¹). Regarding the $\dot{\varepsilon}_{eq}$ range, the $\dot{\varepsilon}_{eq}$ field (not shown) was highly non-uniform across the volume of the food and fluctuated significantly between subsequent

time increments (typical for a Dynamic-Explicit analysis[39]). Although at initial teeth-food contact typical values of $\dot{\varepsilon}_{eq} = 10$ s⁻¹ and $\dot{\varepsilon}_{eq} = 3$ s⁻¹ corresponded to the surfaces and main bulk of the food, respectively. Summarising, for regions where $\varepsilon_{eq} > 0.2$, the applied $\dot{\varepsilon}_{eq}$ range fell again close to the range $0.0001 < \dot{\varepsilon}_{eq} < 5$ at which the viscoplastic-damage law was calibrated. As a result, extrapolating the rate dependent $\sigma_{eq} - \varepsilon_{eq}$ data in order to estimate the constitutive response for $\dot{\varepsilon}_{eq} > 5$ s⁻¹, was not necessary, given also the close experimental-model agreement in Fig. 4a for $\delta = 16.6$ mm s⁻¹.

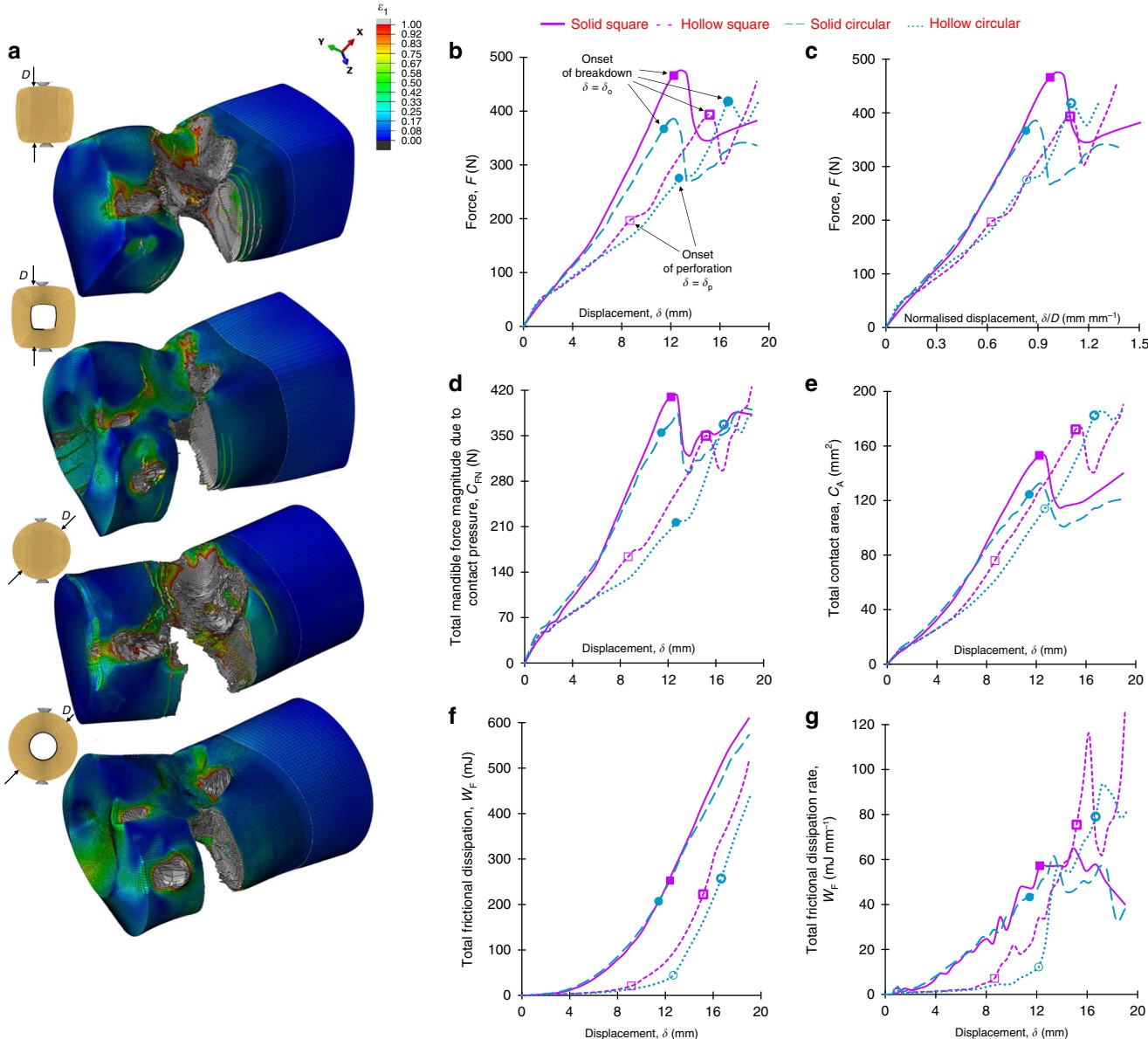

**Fig. 5** Summary of cleaning efficacy Finite Element results. Four food geometric profiles, namely solid square, hollow square, solid circular and hollow circular, are compared in terms of: **a** maximum principal strain contours at maximum bite displacement 19 mm (food cross sectional profiles are shown on the left), **b** force-displacement response, **c** force normalised over the food diameter versus displacement response, **d** total mandible force due to contact pressure versus displacement response, **e** teeth-food total contact surface area versus displacement response, **f** total frictional dissipation versus displacement response and **g** total frictional dissipation rate versus displacement response. In panel **a** from top to bottom the profiles have respective effective diameters 12.6 mm, 13.9 mm, 13.8 mm and 15.2 mm, such that a common cross sectional area of 150 mm² is satisfied

**Teeth cleaning efficacy**. We now demonstrate the effect of food shape on cleaning efficacy by comparing FE predictions of the first bite between four geometric profiles (Fig. 5a), namely: solid square (also utilised in Figs. 3–4), hollow square, solid circular and hollow circular; these have a common cross-sectional area, $A_o \approx 150$ mm², to give rise to a consistent calorie consumption per unit food extrudate length. For hollow profiles, $A_o \approx 150$ mm² can be satisfied by any combination of external fective diameter, $D$, and internal (cavity) effective diameter, the difference of which defines the wall thickness, $Z$; here, a reasonable value of $Z \approx 4.4$ mm is chosen. Figure 5a, b, respectively, compare the deformed food configurations at maximum mandible displacement, $\delta_{max} = 19$ mm and $F - \delta$ data, respectively. Noteworthy is the higher displacement, $\delta$, and particularly larger force required for the onset of breakdown (at $\delta = \delta_o$) in the solid square compared to

the solid circular profile, implying a higher breakdown resistance i.e., area under the $F - \delta$ curve up to $\delta_o$. This owes to the fact that for the same amount of squeezing (common $\delta$), the solid circular profile inherently exhibits larger tensile strains on the external (lateral) food surfaces, compared to the solid square profile. Interestingly, the inverse is found to occur when a cavity is added, such that the hollow circular geometry displays a higher breakdown resistance than the hollow square profile, as well as the highest amongst all profiles. This associates with a profoundly larger $\delta$ required for the lower carnassial (specifically cusp A–see Fig. 4b) to perforate through the section wall (i.e., for fracture to reach the internal cavity surface), the latter being referred as onset of perforation at $\delta = \delta_p$ (denoted in Fig. 5b). Perforation (in the hollow profiles) associates with a marked change between contacting surfaces, as well as overall unloading and redistribution of

strains in the food, reflected by the instant change in the $F - \delta$ slope (see Fig. 5b). These phenomena do not lead directly to complete food separation i.e., splitting into two pieces, and therefore perforation should be distinguished from the onset of breakdown. In fact, perforation and the associated cavity appear to delay ultimate food separation, through increasing overall mechanical energy absorption and by obstructing cracks to propagate across the entire food diameter. As a result of the above, hollow profiles display a less stiff response i.e., lower overall slopes in $F - \delta$ plots, and larger $\delta_o$ than the solid profiles. This, however, encapsulates the effect of inconsistent effective diameter, $D$, between the profiles (see caption in Fig. 5), since increasing $D$ fundamentally increases the $\delta$ required to apply the same strain levels. This effect is eliminated through the $F - \delta/D$ plots in Fig. 5c i.e., upon normalising $\delta$ over $D$, specific to each profile; the latter provides critical information in food design, by isolating the effect of section profile on overall stiffness and breakdown resistance. For example, here Fig. 5c indicates that the hollow sections display different $\delta_o$ in Fig. 5b as a result of inconsistent effective diameter ($D = 15.2$ mm in hollow circular, whereas $D = 13.9$ mm in hollow square) and not, in fact, as a result of different geometry. Interestingly, this is not the case between the solid sections (compare $\delta_o$ difference between solid circular and solid square profiles in Fig. 5b, c). However, the $F - \delta$ response relates to both textural perception and cleaning efficacy, suggesting that both Fig. 5b, c) are important in design.

On the other hand, Fig. 5d–g inform about cleaning efficacy. Figure 5d depicts the magnitude of the resultant force applied through contact pressure on the mandible, $C_{FN}$, versus $\delta$. Notably, the same $C_{FN}$ history applies on the maxilla due to force equilibrium, while based on the simple tangential friction law here[39], the corresponding resultant force due to frictional stress, $C_{FS}$, reads $\mu C_{FN}$. Figure 5d shows that $C_{FN}$ (and thus $C_{FS}$) are proportional to the $F - \delta$ response (Fig. 5a) of each profile, highlighting that a stiffer chewing response increases cleaning efficacy through enhancing overall friction. Nevertheless, this is not always true since high frictional forces may be confined in a small contact region e.g., hard foods tend to breakdown in a brittle manner, with little deformation around the teeth[12,40,41]. In this case, biofilm removal is less likely to occur and therefore $C_{FS}$ plots do not suffice as measures of cleaning efficacy. This gives rise to the total teeth-food contact surface area, $C_A$, plots in Fig. 5e. Importantly, significance is given to the $\delta > 12$ mm regime, since biofilms typically accumulate on the teeth regions close to the gum line[4], which are likely to interact with food at large $\delta$. In this regime ($\delta > 12$ mm), solid profiles display lower $C_A$ than hollow profiles (Fig. 5e). Conclusively, the solid profiles appear less efficacious for biofilm removal. This becomes more clear through the total frictional dissipation energy, $W_F$, versus $\delta$ plots (Fig. 5f) and particularly through the corresponding rate, $\dot{W}_F = dW_F/d\delta$, against $\delta$ plots (Fig. 5g), as these encapsulate both the $C_A$, factors, as well as the amount of sliding between contacting surfaces; the latter is essential to calculate frictional energy[39]. Noteworthy, Fig. 5f can be misleading as $W_F$ are accumulative values, in the sense that frictional energy irrelevant to biofilm removal i.e., when $\delta < 13$ mm (away from the gum line), is embodied to the $W_F$ values at $\delta_{max} = 19$ mm. As a result, solid profiles may appear more efficacious in cleaning than hollow profiles. In fact, for $\delta > 13$ mm the hollow profiles show a considerably higher rate of increase in i.e., larger $\dot{W}_F$, compared to the solid profiles (see Fig. 5g), implying that the largest proportion of frictional dissipation occurs closer to the gum line, raising the potential for biofilm removal.

The above findings are further validated through Fig. 6, where the contact pressure, $c_N$, contours on the teeth at $\delta = 9$ mm and $\delta_{max} = 19$ mm are compared between the solid square (Fig. 6a, b) and hollow square (Fig. 6c, d) profiles; here the corresponding shear frictional stresses are $c_s = \mu c_N$. Owing to the teeth architecture and linear mandible trajectory used here, friction practically occurs in the exterior mandible and interior maxilla surfaces (see Fig. 3e); hence, these surfaces are only examined. In agreement with Fig. 5g at $\delta = 9$ mm, also here the solid square associates with more severe friction than the hollow square profile, based on slightly larger $c_N$ values applied on a slightly larger total teeth surface (compare Fig. 6a, c). Inversely, at $\delta_{max} = 19$ mm, the hollow square profile leads to $c_N$ applied over a profoundly larger total teeth surface (compare Fig. 6b, d).

## Discussion

We have developed an in-silico FE model of oral breakdown in functional pet foods, in order to investigate teeth cleaning efficacy. The need for such advances is increasing due to the severity of oral biofilm accumulation in pets, the large variation in chewing characteristics amongst pets, as well as owing to the complexity and cost associated with in-vitro and in-vivo tests.

The study firstly shows that in-silico modelling of food deformation processes generally requires stress-strain test data collected for strain rate and stress state ranges that cover the corresponding ranges applied in-vivo; otherwise the underlying constitutive law may not be calibrated correctly in foods exhibiting time dependent mechanical behaviour. However, in-vivo rates are very difficult to estimate, since the applied strain rates vary with time during the bite (jaw protrusion)[28], as well as with position in the food. In addition, the strain rates fundamentally scale up with decreasing food item size and/or with increasing bite speed[28]; the latter varies significantly between breeds and can be influenced by several factors[42]. As a result of all the above, the tested strain rate range $0.0001 - 5 \, s^{-1}$, based on which the viscoplastic-damage law was calibrated here, was chosen as large as possible. Testing at $\dot{\varepsilon}_{eq} > 5 \, s^{-1}$ through conventional testing machines was hindered by typical speed limits (here $16.66 \, mm \, s^{-1}$ ($= 1 \, m \, min^{-1}$)). Higher speed testing machines (e.g., hydraulic driven) were not utilised due to typical undesirable noise in the measurements associated with dynamic effects[43], taking also into account the soft nature of the starchy food.

Simultaneously, our results indicate that in order to predict food fracture-damage processes, the fracture toughness and tensile fracture strain parameters are essential. This underlines that tensile test data are also necessary along with compression data, despite that compressive stress states in the food are found clearly dominant during mastication. Our results on starchy extrudates reveal that breakdown initiates due to tensile strains developing on the food lateral surfaces at large teeth indentations. The magnitude of these tensile strains depends largely on Poisson's ratio, $\nu$, suggesting that this property plays an important role in food breakdown and therefore also on teeth cleaning efficacy. For example, a low $\nu$ may allow for large strain food compression, yet with very low associated tensile lateral strains, which in turn decreases the likelihood of breakdown implying more chews per calorie consumption (and potentially enhanced teeth cleaning).

Our mastication FE model is shown to lead to an accurate jaw force-displacement response during the first bite, suggesting that the assumption of an isotropic food constitutive behaviour is a good practise; yet, this may hold specifically for the starch-fibre food system investigated here, which is weak anisotropic owing to the short fibre length and low fibre volume fraction characteristics[33]. The experimental-FE model agreement further suggests that a time independent and stress-state independent unloading food response was also a reasonable simplification; the latter can prove a useful practice in studies where the primary chewing cycle

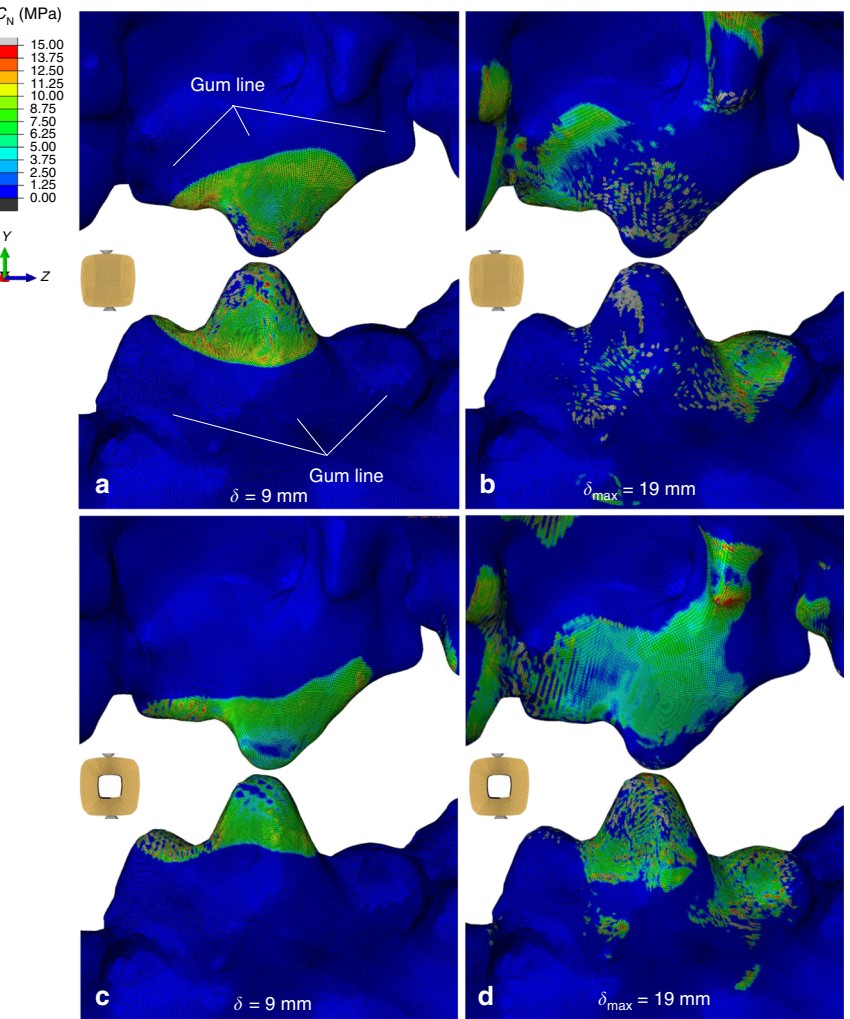

**Fig. 6** Comparison of contact pressure contours. Comparison is performed at mandible displacement 9 mm between (**a**) solid square and (**c**) hollow square profile and at maximum displacement $\delta_{max} = 9$ mm, between (**b**) solid square and (**d**) hollow square profile. Significance is given to contact pressures applied in the gum line regions (denoted) away from the teeth cusps

is important. In this study, the idea of investigating the primary chewing cycle rests on the reasonable assumption that the food extrudate is consumed through a sequence of 'primary' bites across its length, each of them performed on a previously undeformed region[37,40]. It is therefore hypothesised that teeth cleaning efficacy can be assessed based on the first bite response, since also the latter undoubtedly involves a significantly more severe mechanical teeth-food interaction than secondary bites[44,45]. Specifically, although secondary bites typically produce comminuted food particles[28] that likely reach critical teeth regions where biofilms accumulate i.e., gum line[4,6], when this occurs, the associated mechanical tooth-food interaction is not expected strong enough in order for teeth cleaning to take place. Based on all the above, we have developed our FE model to study teeth cleaning within the first bite. We have therefore neglected features associated with food comminution and bolus formation processes i.e., food hydration, disintegration by salivary α-amylase[24,46], as well as soft mouth tissues i.e., gum, tongue[24,25,27,45]. As a result, our model does not currently account for sensory feedback functions e.g., food repositioning by the tongue[45] when a jaw force is exceeded and/or when considerable food-gum contact occurs. Such mechanisms can be readily incorporated in the model, through applying a force driven bite instead of displacement driven bite, and/or via including

a gum layer of a realistic thickness and compliance; different friction coefficients for the gum-food and tooth-food interactions may be also used. Although adding the above features would likely modify the currently presented frictional dissipation rate plots and contact pressure distribution in the gum regions, these results are not expected to change significantly in the teeth regions. Therefore, the main findings of our teeth cleaning efficacy continue to apply.

The highlight of our study is that teeth cleaning efficacy requires the synergy of a large teeth-food contact surface area and high contact pressures. In this regard, the teeth-food friction coefficient may influence teeth cleaning efficacy in different ways; increasing friction coefficient leads towards a stiffer jaw force-displacement response and thus higher pressures, but also towards a more brittle food breakdown behaviour and hence lower total teeth-food contact surface area. We further reveal that hollow food profiles lead to lower contact pressures but higher breakdown resistance, as well as a larger contact surface area compared to solid profiles, eventually enhancing cleaning efficacy.

Our findings open new and exciting possibilities in manufacturing of oral care pet foods, enabled through computer predictions of the effect of several parameters i.e., food texture-size-shape and contacting surface properties, on teeth cleaning efficacy. Our modelling tools are generic such that they

simultaneously inform food design in humans. Specifically, the models can readily be adapted to a range of food systems towards studying consumer preference, as well as the link between food fracturability and aroma release.

## Methods

**Material preparation.** The food was obtained in the form of extrudates in square and rectangular profiles (common cross-sectional area of $A_o \approx 150 \pm 2.5\,\mathrm{mm}^2$), manufactured by Mars Petcare UK. Testing required: (i) sealing the extrudates immediately after production until the experiments and maintaining constant laboratory temperature of 20 °C and relative humidity of 50%, in order to eliminate dehydration, (ii) conducting all tests three weeks following production within three consecutive days, to minimise the effect of ageing (re-crystallisation)[47].

**Mechanical testing.** The in-situ SEM four-point bending (Fig. 1a–c) and uniaxial compression (Fig. 1d, e) specimens were manually cut via razor blades, from the rectangular and square profile extrudates, respectively. The larger scale uniaxial compression (Fig. 1i–l) and tensile (Fig. 1f–h) specimens were cut from the square and rectangular extrudate profiles, respectively, via standard metallic dies. The compression specimen original dimensions (Fig. 1i) satisfy a recommended height over square edge ratio, $L_o/D_o \geq 13/8$, large enough to minimise undesirable sample-platen friction effects on $\sigma_{eq} - \varepsilon_{eq}$ results[48]. Additional care was taken by attaching polyetrafluoroethylene (PTFE) sheets (0.5 mm thickness) on the compression platens and lubricating with silicon oil (0.1 m²/s viscosity). Tension required gluing sandpaper on the grips to eliminate slippage effects while tensile strain was tracked optically within an original gauge length $L_o = 40$ mm marked on the dumbbells (Fig. 1f–h). By assuming incompressibility and uniaxial deformation, the equivalent stress, $\sigma_{eq}$ is calculated as:

$$\sigma_{eq} = \frac{F}{A_i} \qquad (2)$$

, where $F$ and $A_i$ are force and current specimen cross sectional area, while the equivalent strain, $\varepsilon_{eq}$ is determined by:

$$\varepsilon_{eq} = \ln\left(\frac{L_i}{L_o}\right) \qquad (3)$$

for current specimen gauge length, $L_i$ (specimen height for compression).

**Viscoplastic-damage constitutive law.** The coupled viscoplastic-damage law (described in more detail in ref. [49]) implemented in ABAQUS CAE[39] is defined as:

$$\left.\begin{array}{ll} \text{(a) } \sigma_{eq} = E\varepsilon_{eq} & , \sigma_{eq} \leq \sigma_y(\dot{\varepsilon}_{P_{eq}}) \\ \text{(b) } \sigma_{eq} = \sigma_{eq}(\varepsilon_{P_{eq}}, \dot{\varepsilon}_{P_{eq}}, \eta) & , \sigma_{eq} > \sigma_y(\dot{\varepsilon}_{P_{eq}}) \\ \text{(c) } \sigma_{eq} = (1-d)\sigma_{eq}(\varepsilon_{P_{eq}}, \dot{\varepsilon}_{P_{eq}}, \eta) & , \varepsilon_{P_{eq}} \geq \varepsilon_{P_{eq0}}(\dot{\varepsilon}_{P_{eq}}, \eta) \end{array}\right\} \qquad (4)$$

where $\varepsilon_{P_{eq}}$ the equivalent plastic strain (at damage onset $\varepsilon_{P_{eq}} - \varepsilon_{P_{eq0}}$), $\sigma_y$ the yield onset stress, $d$ the scalar damage variable and $\eta$ the triaxiality factor defined through the hydrostatic pressure, $p$, as:

$$\eta = -\frac{p}{\sigma_{eq}} \qquad (5)$$

giving $\eta = -1/3$ in compression, $\eta = 0$ for shear and $\eta = 1/3$ in tension. For $\varepsilon_{P_{eq}} < \varepsilon_{P_{eq0}}(\dot{\varepsilon}_{P_{eq}}, \eta)$, the viscoplastic component is calibrated by converting $\sigma_{eq} - \varepsilon_{eq}$ test data into $\sigma_{eq} - \varepsilon_{P_{eq}}$ via:

$$\varepsilon_{eq} = \varepsilon_{e_{eq}} + \varepsilon e_{P_{eq}} \Leftrightarrow \varepsilon_{P_{eq}} = \varepsilon_{eq} - \frac{\sigma_{eq}}{E} \qquad (6)$$

where $\varepsilon_{e_{eq}}$ the elastic equivalent strain; the $\sigma_{eq} - \varepsilon_{P_{eq}}$ data for each $\dot{\varepsilon}_{P_{eq}}$ are then specified in ABAQUS CAE[39], after assuming $\dot{\varepsilon}_{P_{eq}} \approx \dot{\varepsilon}_{eq}$ based on $E \gg \dot{\sigma}_{eq}$ [49]. For $\varepsilon_{P_{eq}} \geq \varepsilon_{P_{eq0}}(\dot{\varepsilon}_{P_{eq}}, \eta)$, the variable, $d$, is used to determine the stress degradation level by ranging from $d = 0$ (damage onset) up to $d = 1$ (complete damage) with a rate controlled by the fracture energy, $G_f$ (kJ m$^{-2}$), and the characteristic element length, $L_{ch}$. Although $G_f$ is equivalent to the concept of $G_c$, here it is not defined as $G_f = G_c$, as several numerical factors are also taken into account. On the other hand, $L_{ch}$ is defined as the element dimension that sees the maximum $\varepsilon_1$ when $d = 0$[49], which alleviates from numerical errors induced when elements of large aspect ratios are used in the FE mesh[49]. The original formulation ensures a consistent $G_f$ dissipation, independent of element size and equivalent to the area under an equivalent stress-displacement $\sigma_{eq} - u$, degradation response, defined as:

$$G_f = \int_{u0}^{uf} \sigma_{eq} du \qquad (7)$$

where $u_0$, $u_f$ are, respectively, the element displacements at $d = 0$ and $d = 1$

(corresponding to equivalent strains $\varepsilon_{eq0}$ and $\varepsilon_{eqf}$), calculated through:

$$u = L_{ch}\varepsilon_{eq} \qquad (8)$$

For damage onset ($d > 0$) to occur it is firstly required that $\eta \geq 0$ such that degradation only applies in shear, tension, as well as any multiaxial tensile state[49]. Secondly, the maximum principal strain, $\varepsilon_1$, must exceed the experimental $\varepsilon_f(\dot{\varepsilon}_{eq})$ values (Eq. (1)). Within $0 \leq \eta \leq 1/3$, the influence of compressive stress components in the $G_f$ calculation via Eq. (7) (these occur for example when $\eta = 0$) is rectified by replacing $\sigma_{eq}$ and $\varepsilon_{eq}$ by $\sigma_1$ (maximum principal stress) and $\varepsilon_1$, respectively; these correspond to $\sigma_{10}$, $\varepsilon_{10}$ at $d = 0$, and $\sigma_{1F} = \varepsilon_F = 0$ at $d = 1$, which are used to define $R_\sigma$, $R_\varepsilon$ as:

$$R_\sigma = \frac{\sigma_{eq0}}{\sigma_{1_0}} \qquad (9)$$

$$R_\varepsilon = \frac{\varepsilon_{eq0}}{\varepsilon_{1_0}} = \frac{\varepsilon_{eqf}}{\varepsilon_F} \qquad (10)$$

to finally define $G_f$ as:

$$G_f = R_\sigma R_\varepsilon \frac{G_c}{2} \qquad (11)$$

The factor of $\frac{1}{2}$ (50% reduction) is applied to $G_c$ to account for the fact that the current viscoplastic-damage law predicts crack propagation in the form of two adjacent layers of elements undergoing damage[49]. $G_c = 0$ kJ m$^{-2}$ is used for elements of external food surfaces in the mastication model, in order to enforce element deletion (crack initiation) as soon as $d > 0$; otherwise, due to no initial crack in the food item, $d > 0$ does not guarantee strain localisation, and thus $G_f$ is likely not be dissipated correctly i.e., along a single crack[49]. For the rest of the food mesh elements, the true $G_c = 0.93$ kJ m$^{-2}$ is used in Eq. (11), as previously described.

**Oral processing FE model construction.** The boundary conditions of the FE model involve fixed maxilla and mandible, the latter being only enforced to translate along the $Y$-axis (see Fig. 3) at various rates, $\dot{\delta}$, until teeth-teeth contact is reached at $\delta_{max} = 19$ mm. The virtual teeth geometry (Fig. 3a) is constructed as follows: (i) 3D Computed Tomography (CT) skull data under centric relation state[36,50] are obtained (Creaform EXAscan instrument $-0.05$ mm voxel size) from a domestic boxer donated to research under the owner's consent, (ii) the CT data are converted via RAPIDFORMXOR into a Computed Aided Engineering (CAD) shell geometry (Fig. 3a) and lastly (iii) the CAD geometry is imported into ABAQUS CAE and cropped (dashed box–Fig. 3a) into three mandibular and four maxillary teeth (Fig. 3b, c). The virtual food item (Fig. 3d, e) is constructed through volume extrusion of the exact cross-sectional profile of one square extrudate specimen, captured by optical microscopy.

Teeth and food are respectively meshed in ABAQUS CAE[39] by 155079 4-node bilinear rigid elements (minimum element dimension 0.07 mm), and 932564 3D stress 8-node linear brick reduced integration enhanced hourglass control elements (minimum element dimension $T = 0.05$ mm)[39]. Tie constraint is enforced between the coarse food mesh region (minor strains) and fine mesh region (large strains and fracture) (see Fig. 3e); in the latter, original element aspect ratios, $A = H/W \approx 3$, (see Fig. 3f, g) are used to minimise overall mesh distortion[11,51] i.e., upon teeth indentation the compressive strains act along the long element dimension, $H$ (height); adaptive meshing strategies towards improving the performance of the mesh at large teeth indentations e.g., the ABAQUS arbitrary Lagrangian-Eulerian (ALE) technique[39], are found to increase prohibitively computational cost in this large model. The ABAQUS[39] Dynamic-Explicit analysis is run via a conventional workstation (Intel(R) Core(TM) i7 3770 CPU 3.4 GHz) with semi-automatic mass scaling based on target time increment, $\Delta t = 10^{-5}$ s, and specified true food material density, $\rho = 1410$ kg m$^{-3}$ [49]; these parameters ensured a quasi-static response i.e., total kinetic energy less than 2% of total energy associated with food deformation and fracture[39]. Since perfect incompressibility i.e., Poisson's ratio, $\nu = 0.5$ cannot be enforced within an FE Explicit analysis[39], an approximate value of $\nu = 0.475$ is used; the latter is a good compromise on the basis that the material is not highly confined[39], i.e., the total free surface to total volume ratio remains small during the bite. General contact is employed, allowing for all facets of all the food elements to potentially interact with the teeth surfaces, as well as between themselves. This accounts for new element facets being exposed as 'fracture' in the form of element deletion evolves. A surface geometry correction algorithm[39] is also enforced to reduce contact noise by automatically smoothing surfaces with discontinuous face normals. The analysis is run parametrically for five friction coefficients: $\mu = 0$, 0.15, 0.3, 0.5, 0.8, associated with the tangential frictional behaviour, while a 'hard' pressure-overclosure relationship[39] is consistently used for the interaction along the normal direction. These conditions are enforced within the classical isotropic Coulomb friction law[39], assuming that $\mu$ is independent of slip rate and contact pressure.

## Data availability

The raw/processed data required to reproduce these findings can be made available upon reasonable request from the authors.

## Code availability

The computer code required to reproduce these findings can be available upon reasonable request from the authors.

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

## Acknowledgements

Mars Petcare is acknowledged for the financial support and for providing the material. Stephen Johnson is also acknowledged for producing the 3D printed teeth geometries and Dr. Ruth Brooker for providing training on the SEM and mechanical testing equipment.

## Author contributions

C.G.S. conceived the study under the supervision of M.N.C. M.E. produced the extruded material, provided feedback throughout the study and comments on the paper and gave the final approval for publication.

## Additional information

**Competing interests:** The authors declare no competing interests.

