## [Peer Review File · Nature Communications]

Reviewers' comments:

Reviewer #1 (Remarks to the Author):

SUMMARY

The authors propose a 3D finite element (FE) model of canine chewing (applied to a specific pet food) to study biofilm accumulation and its negative effects on dental health. First, standard tension and compression tests are performed on samples of the pet food to understand both force-displacement response and the reaction of the food microstructure to fracturing. Second, a CT scan of a dog's head was used to develop a 3D FE model of the teeth, jaw and skull and also to produce a 3D printed stainless steel physical model. Third, matching simulations and experiments of singular biting were performed to evaluate the FE model response. Good agreement was observed between force-displacement response and fracture behaviour of the simulations and experiments. Fourth, the effect of changes to food pellet shape on breakdown behaviour and area of contact with the teeth and gums is predicted. The authors propose the use of the FE model for evaluating the ability for pet food to clean teeth and to remove biofilms.

REVIEW SUMMARY

The paper constitutes a valuable scientific contribution and is very well written and structured. I have two major comments and a few minor comments that need to be addressed before I'd recommend publication.

MAJOR COMMENTS

- A large component of the FE model development is focused around the time dependent response of material model. The authors are critical of previously published models for "ignoring time dependent effects" and the results of Figure 2 show material model calibrations across a wide range of strain rates. However, the model is applied for only one loading rate (16 mm/s; Figure 4). The authors identify that the other loading rates used in the experiments (0.016 mm/s, 0.16 mm/s, 1.6 mm/s) are not representative of normal chewing. A better approach for Section 4 (in my opinion) would have been to identify the range of loading rates that apply to chewing and to run the model across these ranges of those loading rates. I suggest that the authors repeat experiments and simulations at a more appropriate range of loading rates to prove the validity and necessity of using a time-dependent material model in the mastication application.
- It is not established if the FE model results are dependent on the choice of element shape and size. The FE method for continuum solid mechanics involves the discretisation of solid matter into elements with pre-defined shapes. There is uncertainty in how much error in model outputs is introduced by this discretisation. Two tests are required: (1) Does the mesh spatial resolution affect the model results? (2) Does the element shape and initial orientation affect the model outputs? The authors mention that results have a "small mesh dependency [32]" but this was evaluated for a different type of deformation and may not be applicable for the mastication application. This needs to be specifically tested and reported on. The direction of cracking in the simulation appears to match the crack directions in the experiment quite well. However, this direction is also approximately aligned with the initial plane of element edges and faces. The authors mention that "Adaptive meshing strategies e.g. arbitrary Lagrangian-Eulerian [33] are found to increase prohibitively computational cost in this large model." This comment points to the need for predicting arbitrary crack directions that do not necessarily align with element faces. The authors need to test if the choice of element size, type and orientation affects the model results and present the outcomes of this analysis.

MINOR COMMENTS

- Equation 2 is define as a function of strain rate, not strain. Should the term in the time integral be

strain rate instead of strain?

- In Section 2 the authors write “Figure 1(c) highlights a large extend of fibre-bridging during fracture upon bending, underlining the importance of G_c in describing the significant energy dissipation required to create new fracture surfaces [29] in this material”. Two comments: (1) the word “extend” appears wrong. Should this be “extent”? (2) How, specifically does the fibre-bridging during bending type fracturing underline the importance of using G_c ? This should be explained here in a short sentence.
- Does the model include soft tissues such as gums? Or only bone and teeth? If the latter is the case then this needs to be made clear in the results for Section 6 (including Figure 6). What affect would including compliant gum tissue have on results shown in Figure 6? This should be discussed.
- End of section 6. There is no “Figure 13”. Should this be Figure 6?

Reviewer #2 (Remarks to the Author):

This paper deals with computer simulation of food oral processing by applying finite element method. The reviewer thinks this approach by using FEM is sound and correct. The authors are applying solid mechanics approach to this problem. The authors discuss cleaning efficacy but the reviewer cannot judge the validity and usefulness. If another reviewer in such field agrees with the validity and the usefulness, this paper is suitable for publication.

Reviewer #3 (Remarks to the Author):

This manuscript proposed a novel approach to study the teeth cleaning efficacy in pet through the combined knowledge of fundamental material properties characterisations (both mechanical and fracture properties), constitutive material model and computational simulation using finite element technique. This is very interesting work that attempts to tackle such a complex problem on oral processing/chewing characteristics in pets. The research in this area is still scarce and deserves more attention, not only for the needs to enhance knowledge but also for direct manufacturing of oral care pet foods in pet's industry. In this work, however, the authors have made several assumptions to simplify the problem and these assumptions have significant effect on the results presented in this work.

1. The assumption of incompressibility is intriguing since the extrudates shown in Figure. 1(b) reveals a fibrous structure. The composition of these extrudates contains about 2.5% cellulose fibres. Should the presence of fibres violate the assumption of incompressibility? Can you confirm this? Also, if these are long fibres, it would affect the mechanical response depending on the loading deformation with respect to the fibre direction. However, the constitutive model used in this study did not account for this material anisotropy. This should be clarified.

2. Although additional care was taken to minimise the friction between the sample and compression platens, it appears from Figure 1(e) that friction is completely eliminated; sample shows a barrel shape and sample fractured in compressive direction, rather than the maximum shear plane.

3. In Figure 2, the proposed constitutive material model described both the compression and tension behavior of the sample, but not the unloading behavior. Especially, less satisfactory agreement is observed for unloading in tension compared to that in the compression mode. This may suggest the large extent of impact from the presence of cellulose fibre in the extrudates.

4. The mastication model was simplified by excluding the effect of saliva. However, saliva contains amylase enzyme which can readily digest starchy foods such as the extrudates studied here, thus affecting its deformation behavior to a certain extent. However, the material data does not account for

this affect so the mastication model is considered rather simplified. In addition, the saliva may have large effect on the stickiness and in turn the frictional behavior between the sample and both the maxilla and mandible too.

5. The mastication model used in this work focuses mainly on the first bite. According to the work of Stokes et al. (2013) in Current Opinion in Colloid & Interface Science 18, 349–359, oral processing (although in human) is a dynamic process, involving first bite, chewing (comminution), granulation, bolus formation and swallowing. Accordingly, not only the first bite but the chewing/comminution process which create smaller/broken extrudates should contribute to the cleaning efficacy too. However, the mastication model did not capture this, either due to the constitutive model used or the limited amount of chewing cycles prescribed in the FE model. This deserves some comment, perhaps how the current model can be improved and whether how those broken/fractured extrudates may help/contribute to the cleaning efficacy as they should be smaller enough and easier to get to the region near the gum line.

6. How does the assumed displacement rate of 16.6 mm/s relate to the typical chewing speed of dogs/pets? With this rate, the strain rate of the extrudate would increase as D ($D = 12.6$ mm at time 0) decreases. So, at higher mandible displacement, the strain rate can then exceed the range of experiment data, ,maximum strain rates of 5/s given in Figure 2(a) for compression and 1/s given in Figure 2(d) for tension. How did the FE model extrapolate for the deformation out of this range?

7. What is the friction law used in the FE model? Is it a simple Coulomb's law? Any critical shear stress before the friction is activated?

8. In Explicit, the simulation results, especially the fracture model, is highly dependent on the mesh size. How is this optimised in the model?

9. Even though the displacement rate of 1.66 mm/s with $\Delta t = 10$ s are not realistic conditions for mastication (based on the literature values reviewed by the authors), the effectiveness and predictive capability of the proposed FE model can be analysed by plotting the predicted mandible force to compare with the experimental data shown in Figure 4(a). By doing so, the assumption of coefficient of friction of 0.3 can also be verified.

Other comments

1. There are few typos. The whole manuscript should be rechecked.

- Page 3, "a large extend" should be "a large extent"

- Page 9, " $\Delta t_{t} = 1^{-5}$ s" should be " $\Delta t_{t} = 10^{-5}$ s"

2. Page 14, there is no Figure 5(h).

3. Page 15, there is no Figure 5(h), this should be Figure 5(g). Also, there is no Figure 13, this should be Figure 6.

RESPONSE TO REVIEWERS

Manuscript number: NCOMMS-18-35463-T

‘Computer simulations of food oral processing to engineer teeth cleaning’

07/05/2019

Dear editor and reviewers,

We would like to thank you for your interest in our work and moreover for your beneficial comments.

We have responded to all the comments and provided additional numerical data as detailed below. The corresponding changes in the revised manuscript are highlighted.

Reviewer 1

Major Comment 1: A large component of the FE model development is focused around the time dependent response of material model. The authors are critical of previously published models for “ignoring time dependent effects” and the results of Figure 2 show material model calibrations across a wide range of strain rates. However, the model is applied for only one loading rate (16 mm/s; Figure 4). The authors identify that the other loading rates used in the experiments (0.016 mm/s, 0.16 mm/s, 1.6 mm/s) are not representative of normal chewing. A better approach for Section 4 (in my opinion) would have been to identify the range of loading rates that apply to chewing and to run the model across these ranges of those loading rates. I suggest that the authors repeat experiments and simulations at a more appropriate range of loading rates to prove the validity and necessity of using a time-dependent material model in the mastication application.

Reply: The authors recognise the above concerns and discuss the issue of rate dependency extensively in the revised manuscript. This demonstrates the capability of the model in capturing rate dependent effects. Before referring to the corresponding changes in the manuscript, the authors would like to underline that even for cases when a single chewing/loading rate (mm/s) is studied, it is not reliable to use a food constitutive law calibrated from tests at a single strain rate (1/s) (for rate dependent materials). This is because a single chewing/loading rate involves varying strain rates during the duration of the bite, which are also extremely difficult to estimate a-priori i.e. without simulations. As a result, an ideal strain rate for which tests should be performed cannot be easily found.

Relevant text is firstly added in Section 2:

'The tested strain rate range 0.0001–5/s is chosen as large as possible in order to: (i) investigate the extent of material time dependency, and (ii) span the actual strain rates applied during chewing, such that the food constitutive law is calibrated correctly (based on these rates). As this study shows, the applied strain rate field, $\dot{\epsilon}_{eq}(x, y, z, t)$, varies with time, t , during the bite (jaw protrusion), and with position (x, y, z) in the food item. In addition, $\dot{\epsilon}_{eq}(x, y, z, t)$ fundamentally scales up with decreasing food item size and/or with increasing bite speed; the latter varies significantly between breeds and can be influenced by several factors (to be discussed in Section 5). The above suggest that the in-vivo rates are very difficult to estimate and that one cannot choose a single characteristic rate to obtain material data and subsequently use a time independent constitutive law, particularly in the foods investigated here. Instead, obtaining data for varying rates within a wide rate range appears to be the best practise. Testing at $\dot{\epsilon}_{eq} > 5/s$ through conventional testing machines was hindered by typical speed limits (here 16.66 mm/s (=1 m/min)). High speed machines (e.g. hydraulic) were not utilised due to typical undesirable noise in the measurements associated with dynamic effects [33], taking also into account the soft nature of the materials investigated here.'

Thereafter, Section 5 now includes:

'The jaw speed range 0.016– 16.6 mm/s is chosen as large as possible in order to: (a) investigate time dependency, and, (b) cover the bite speed range applied in-vivo. For the latter, relevant literature is currently poor, and whether/how food texture and relative food item size relates to the chewing rates in mammals and particularly in domestic dogs, is debatable and has not been yet fully understood. On average between pet breeds, Gerstner et al [41] reported a chewing cycle period, $CD = 0.41$ s, defined as the time between consecutive jaw openings. By assuming that 70% of CD is consumed for the actual biting action i.e. a bite duration, $BD \approx 0.3$ s, and a total jaw displacement of 19 mm for teeth contact (based on the configuration shown in Figures 3(d-e)), a characteristic bite speed of, $\dot{\delta}_{ch} = 63.33$ mm/s, is determined. Unfortunately, applying $\dot{\delta} > 16.6$ mm/s consisted an experimental challenge (see also Section 2), as it required high speed testing machines together with an additional fast feedback mechanism (e.g. magnetic grips) that prevents contact between the jaws. Nevertheless, $\dot{\delta}_{ch}$ should be regarded here as a rough estimate of the upper bound for the bite speed, as we have assumed that the mandible translates until teeth contact occurs, independent of sensory feedback i.e. associated forces; note also that the required translation of 19 mm is specific to the food item diameter/height, $D = 12.6$ mm and configuration shown in Figures 3(d-e). As a result, the fact that experimental data are only collected up to $\dot{\delta} = 16.6$ mm/s, does not hinder our mastication model development as well as cleaning efficacy investigation, but it could be of a concern in detailed studies of texture perception. Based on the above, the following analysis (including cleaning efficacy – Section 6) focuses on experimental-model results for $\dot{\delta} = 16.6$ mm/s; however, for verification purposes simulations are also performed for $\dot{\delta} = 1.66$ mm/s and $\dot{\delta} = 166.6$ mm/s (results reported in Appendix–A).'

Finally, the effect of chewing speed on the chewing response is analysed in Appendix-B.

Major Comment 2: It is not established if the FE model results are dependent on the choice of element shape and size. The FE method for continuum solid mechanics involves the discretisation of solid matter into elements with pre-defined shapes. There is uncertainty in how much error in model outputs is introduced by this discretisation. Two tests are required: (1) Does the mesh spatial resolution affect the model results? (2) Does the element shape and initial orientation affect the model outputs? The authors mention that results have a “small mesh dependency [32]” but this was evaluated for a different type of deformation and may not be applicable for the mastication application. This needs to be specifically tested and reported on. The direction of cracking in the simulation appears to match the crack directions in the experiment quite well. However, this direction is also approximately aligned with the initial plane of element edges and faces. The authors mention that “Adaptive meshing strategies e.g. arbitrary Lagrangian-Eulerian [33] are found to increase prohibitively computational cost in this large model.” This comment points to the need for predicting arbitrary crack directions that do not necessarily align with element faces. The authors need to test if the choice of element size, type and orientation affects the model results and present the outcomes of this analysis.

Reply: The authors recognise the reviewer’s concerns and have now presented a rigorous mesh sensitivity analysis and discussion in Appendix-B. The analysis is performed both via using the classical fracture energy definition (Equation (11)), as well as via using a correction strategy against mesh dependent effects. The authors would also like to clarify that “Adaptive meshing strategies” is mentioned as a way of improving the performance of the mesh at large teeth indentations i.e. large strains, and not as a technique necessary for predicting arbitrary cracks; this is also clarified in the revised manuscript. In addition, further to the discussion included in Appendix-A regarding potential mesh orientation effects, here the authors also provide an example where the major crack faces of the food do not coincide with the original plane of the element facets. This is demonstrated via Figure R1 below, which corresponds to conditions: $T = L_{ch} = 0.05$ mm, $H/W = 0.3/0.1$, $\dot{\delta} = 16.6$ mm/s and $\mu = 0$. Although, some alignment (between crack direction and original plane element facets) can be seen for the primary stages of crack propagation (due to the reasons explained in Appendix-B), thereafter the crack path changes as the crack propagates further towards the core of the food item.

Figure R1. Detail of the major crack face morphology.

Finally, the authors would like to highlight that the currently used characteristic element length, L_{ch} , definition based on the element dimension that undergoes the maximum elongation, has been shown to correct from potential errors in the damage computations i.e. the integration of G_f , induced by large element aspect ratios and element orientation effects [1]. Note also that if a mesh orientation different to the one shown in the manuscript in Figures 3(f)&(g) is used, e.g. if the elements are oriented such that their height, H , is inclined to the direction of teeth indentation (direction of Y axis), this would compromise the performance of the mesh at large indentations.

Minor Comment 1: Equation 2 is defined as a function of strain rate, not strain. Should the term in the time integral be strain rate instead of strain?

Reply: The authors identify the confusion caused by Equation (3), due to the fact that the tensile failure strain is presented as a function of strain rate, $\varepsilon_f(\dot{\varepsilon}_{eq})$, while the term in the integral was strain. In fact, the term in the integral is strain history, $\varepsilon_{eq}(t)$; the latter depends on the associated strain rate and this is how strain rate dependency is induced. For clarity now the revised manuscript includes the following text:
'This is supported by the strong relationship ($R^2 = 0.97$) found between the monotonic $\varepsilon_f(\dot{\varepsilon}_{eq})$ data obtained from Figure 2(d) and the integration of strain history, $\varepsilon_{eq}(t) = \dot{\varepsilon}_{eq}t$, over time:

$$\varepsilon_f(\dot{\varepsilon}_{eq}) = -0.024 \ln \left(\int_0^{t_f} \dot{\varepsilon}_{eq} t \, dt \right) + 0.2 \quad (3)$$

where t_f is time elapsed from $\varepsilon_{eq} = 0$ until failure ($\varepsilon_{eq} = \varepsilon_f$); note that for non-monotonic loading a general strain history $\varepsilon_{eq}(t)$ must be integrated in Equation (3), instead of $\dot{\varepsilon}_{eq}t$, and this is utilised to define the tensile strain threshold value in the damage onset criterion of the material constitutive law.'

Minor Comment 2: In Section 2 the authors write “Figure 1(c) highlights a large extend of fibre-bridging during fracture upon bending, underlining the importance of G_c in describing the significant energy dissipation required to create new fracture surfaces [29] in this material”. Two comments: (1) the word “extend” appears wrong. Should this be “extent”? (2) How, specifically does the fibre-bridging during bending type fracturing underline the importance of using G_c ? This should be explained here in a short sentence.

Reply: (1) The authors have corrected the typographic error. (2) We realise that our previous statement was not clear enough. Also fibre bridging occurs in any tensile deformation and therefore it should not be necessarily linked with bending. We have now modified the text and explained further as such:

'Figure 1(c) shows that crack propagation involves significant fibre-bridging, a mechanism which has been reported [29] to enhance fracture toughness, G_c (local energy dissipation required to create new fracture surfaces) [30] in this material; the latter added to the tough nature of the starch matrix, clearly suggests that G_c is an important parameter in describing damage/fracture in this material.'

Minor Comment 3: Does the model include soft tissues such as gums? Or only bone and teeth? If the latter is the case then this needs to be made clear in the results for Section 6 (including Figure 6). What affect would including compliant gum tissue have on results shown in Figure 6? This should be discussed.

Reply: The authors fully agree that some discussion should be dedicated to the above aspect. This is now addressed in the revised manuscript via adding the following paragraph (section 6):

'These above results lend weight to the analysis, particularly to the newly introduced concept of \dot{W}_F as a comprehensive measure of cleaning efficacy, based on which different food geometries and textures can be compared in a straightforward manner. On the other hand, the c_N distribution is a more detailed measure which identifies sites on the teeth with high and low potential for biofilm removal. As mentioned earlier (Section 5), a more sophisticated approach would be to define the gum line, such that one can focus on c_N values applied specifically on the respective regions. This may also involve modelling a realistic thickness and compliance for the gum layer, as well as a different friction coefficient, μ , for the gum-food contact, compared to the $\mu = 0.3$

used for tooth-food contact. Although this could potentially modify the \dot{W}_F results at large bite displacements as well as the c_N distribution on the gum regions, yet this would not alter the results of this cleaning efficacy study, since the first bite breakdown response and corresponding c_N distribution on the teeth, are not expected to change considerably.'

In addition, an relevant introductory paragraph is also structured in Section 4, in order to explain in detail the assumptions/simplifications of our model (which relate to gum tissues):

'As long as the mechanics of the first bite are considered, it is reasonable for our current model to neglect features associated with food comminution and bolus formation processes [22, 40] i.e. salivary effects (food hydration and disintegration by α -amylase) [22], as well as soft mouth tissues (gum, food repositioning by the tongue) [22, 23, 25]. Although comminuted food particles (as a result of several bites) can potentially reach teeth regions close to the gum line, where biofilms typically accumulate [4, 6, 10], this may not contribute to teeth cleaning. This is because for the foods considered here biofilm removal does occur chemically, but instead it requires a certain degree of mechanical tooth-food interaction (friction). Conclusively, the main objectives here are the in-depth understanding and modelling of the mechanical food breakdown and associated first-bite tooth-food interaction, based on a solid mechanics approach.'

Minor Comment 4: End of section 6. There is no "Figure 13". Should this be Figure 6?

Reply: The authors apologise. The error has been corrected.

Reviewer 2

Comment: This paper deals with computer simulation of food oral processing by applying finite element method. The reviewer thinks this approach by using FEM is sound and correct. The authors are applying solid mechanics approach to this problem. The authors discuss cleaning efficacy but the reviewer cannot judge the validity and usefulness. If another reviewer in such field agrees with the validity and the usefulness, this paper is suitable for publication.

Reply: -

Reviewer 3

Comment 1: The assumption of incompressibility is intriguing since the extrudates shown in Figure 1(b) reveals a fibrous structure. The composition of these extrudates contains about 2.5% cellulose fibres. Should the presence of fibres violate the assumption of incompressibility? Can you confirm this? Also, if these are long fibres, it would affect the mechanical response depending on the loading deformation with respect to the fibre direction. However, the constitutive model used in this study did not account for this material anisotropy. This should be clarified.

Reply: The authors recognise the reviewer's concerns related to the assumption of incompressibility and the associated role of fibres and address this through adding the following text (Section 2):

'Incompressibility is assumed based on Poisson's ratio, $\nu = 0.5$, previously determined in a similar starch-fibre recipe, both for compression and tension [20]. It has been also identified that the degree of anisotropy due to potential fibre orientation effects induced by extrusion, is small in terms of stress-strain response; this owes to the short fibre length and low fibre volume fraction characteristics. Conclusively, an isotropic constitutive law based on uni-directional experimental data is considered here to be good representation of the starchy food material response.'

Comment 2: Although additional care was taken to minimise the friction between the sample and compression platens, it appears from Figure 1(e) that friction is completely eliminated; sample shows a barrel shape and sample fractured in compressive direction, rather than the maximum shear plane.

Reply: The authors agree with the above statements, but also would like to clarify further the experimental observations through the following revised text (Section 2):

'Figure 1(b) shows that failure is triggered by the initiation and growth of micro-cracks occurring in the volume of the specimen under tensile deformation (top surface); these micro-cracks are perpendicular to the direction of tensile strain, which agrees with Figure 1(e), where during compression excessive micro-cracking occurs along the direction of compressive loading and normal to the lateral tensile strain induced by associated specimen expansion effects.'

We would also like to emphasise that when significant barrelling occurs the corresponding data are not regarded valid, as now clarified by the following text (Section 2):

'The compression $\sigma_{eq} - \varepsilon_{eq}$ curves (Figure 2(a)) for each $\dot{\varepsilon}_{eq}$, are plotted up to the point where Equations (1-2) hold; beyond this point (which varies with $\dot{\varepsilon}_{eq}$), significant specimen barrelling and micro-cracking effects (similar to the ones in Figures 1(e), 1(k-l)) initiate.'

Comment 3: In Figure 2, the proposed constitutive material model described both the compression and tension behavior of the sample, but not the unloading behavior. Especially, less satisfactory agreement is observed for unloading in tension compared to that in the compression mode. This may suggest the large extent of impact from the presence of cellulose fibre in the extrudates.

Reply: The authors recognise the need for interpreting the above model-experimental mismatch. This is now addressed via adding the following paragraph (Section 3):

'The experimental-model agreement (Figure 2) was achieved by fitting an initial small elastic regime governed by elastic modulus, $E = 50$ MPa, followed by a time and stress state dependent elastoplastic regime, which can also degrade to capture damage. This viscoplastic representation gives an excellent experimental-model fit in terms of the monotonic response (Figures 2(a)&(d)) and reasonable model predictions for the stress relaxation behaviour (Figure 2(c)&(f)). Some discrepancy however exists in the unloading response (particularly in tension – Figures 2(b)&(e)), primarily attributed to the model's assumption of both time independent and stress state independent elastic unloading determined by $E = 50$ MPa, which does not account for the fact that the tensile response is generally stiffer than the compressive i.e. unloading from the same absolute maximum strain begins from higher stresses in tension than in compression. Whether tension also involves larger cyclic hysteresis due to a potentially larger extent of damage mechanisms e.g. micro-cracking, fibre-matrix debonding, remains to be investigated. Nevertheless, extending the constitutive law in order to capture the actual time dependent and stress state dependent unloading response of the food was not considered necessary in the current study where a single chewing cycle is investigated.'

Comment 4: The mastication model was simplified by excluding the effect of saliva. However, saliva contains amylase enzyme which can readily digest starchy foods such as the extrudates studied here, thus affecting its deformation behavior to a certain extent. However, the material data does not account for this affect so the mastication model is considered rather simplified. In addition, the saliva may have large effect on the stickiness and in turn the frictional behavior between the sample and both the maxilla and mandible too.

Reply: The authors are aware of the profound effect of saliva on starch destructurement as well as other oral processing attributes and acknowledge that the reasons for not including such phenomena in our model needs to be reported. Before referring to the corresponding added text in the revised manuscript, we would also like to emphasise that progressed stages of oral processing (where also salivary effects become increasingly dominant over mechanical breakdown) involve significant chemical breakdown, including food transformation from a solid into a liquid state, such that a solid mechanics approach is no longer be valid but instead an advanced rheological computational modelling approach remains to be developed. The latter consists a longstanding challenge, related both to the construction and experimental calibration of the food constitutive law (which would need to account for degradation of mechanical properties both due to mechanical loading and due to chemical activity

and/or hydration), but also to the implementation and validation of such a complex numerical analysis. Prior to addressing the above tasks, mechanical food breakdown and the associated primary tooth-food interaction needed to be understood first, and this would be hindered if certain simplifications were not made.

We have therefore added the following paragraph in Section 4 (which also addressed the reviewer's next comment):

'As long as the mechanics of the first bite are considered, it is reasonable for our current model to neglect features associated with food comminution and bolus formation processes [22, 40] i.e. salivary effects (food hydration and disintegration by α -amylase) [22], as well as soft mouth tissues (gum, food repositioning by the tongue) [22, 23, 25]. Although comminuted food particles (as a result of several bites) can potentially reach teeth regions close to the gum line, where biofilms typically accumulate [4, 6, 10], this may not contribute to teeth cleaning. This is because for the foods considered here biofilm removal does occur chemically, but instead it requires a certain degree of mechanical tooth-food interaction (friction). Conclusively, the main objectives here are the in-depth understanding and modelling of the mechanical food breakdown and associated first-bite tooth-food interaction, based on a solid mechanics approach.'

We would also like to underline our initial statements in Section 4 (from previous version of manuscript), which support further the argument that studying the first bite is a good and valid approach for the current teeth cleaning application. These are:

'We focus on phenomena dominant in the primary chewing cycle i.e. carnassial teeth indentation under dry conditions on food material associated with zero strain history [35, 36]. This rests on the reasonable assumption that the food extrudate is consumed through a sequence of 'primary' bites across its length, each of them performed on a previously undeformed food region.'

Comment 5: The mastication model used in this work focuses mainly on the first bite. According to the work of Stokes et al. (2013) in Current Opinion in Colloid & Interface Science 18, 349–359, oral processing (although in human) is a dynamic process, involving first bite, chewing (comminution), granulation, bolus formation and swallowing. Accordingly, not only the first bite but the chewing/comminution process which create smaller/broken extrudates should contribute to the cleaning efficacy too. However, the mastication model did not capture this, either due to the constitutive model used or the limited amount of chewing cycles prescribed in the FE model. This deserves some comment, perhaps how the current model can be improved and whether how those broken/fractured extrudates may help/contribute to the cleaning efficacy as they should be smaller enough and easier to get to the region near the gum line.

Reply: The authors agree with the above statements and ideas and have already addressed these through the added paragraph presented above in reply to the previous comment (comment 4). In addition, the work by Stokes et al. (2013) is also reviewed and referenced in the text.

Comment 6: How does the assumed displacement rate of 16.6 mm/s relate to the typical chewing speed of dogs/pets? With this rate, the strain rate of the extrudate would increase as D (D = 12.6 mm at time 0) decreases. So, at higher mandible displacement, the strain rate can then exceed the range of experiment data, maximum strain rates of 5/s given in Figure 2(a) for compression and 1/s given in Figure 2(d) for tension. How did the FE model extrapolate for the deformation out of this range?

Reply: The authors agree with the above statements. Relevant concerns are also raised by the first reviewer, and we have generally recognised that the aspect of material rate dependency in relation to chewing speed has not been studied closely before and therefore we ought to provide more analysis and discussion. A separate Section in the Appendix-B is thus also now provided with additional data related to rate dependency.

However, we would like to point out that extrapolating material data for strain rates outside the tested range was not found to be necessary here, as stated now in Section 5 by:

'Summarising, for regions where $\varepsilon_{eq} > 0.2$, the applied $\dot{\varepsilon}_{eq}$ range fell again close to the range $0.0001 < \eta < 5$ at which the food constitutive law was calibrated (Section 2). As a result, extrapolating the rate dependent $\sigma_{eq} - \varepsilon_{eq}$ data in order to estimate the constitutive response for $\dot{\varepsilon}_{eq} > 5/s$, was not necessary, given also the close experimental-model agreement in Figure 4(a) for $\delta = 16.6$ mm/s.'

Regarding the experimental rates, the following text is added (Section 2):

'The tested strain rate range 0.0001–5/s is chosen as large as possible in order to: (i) investigate the extent of material time dependency, and (ii) span the actual rates applied during chewing in-vivo, such that the underlying food constitutive law is calibrated correctly based on these rates; these, however, are difficult to estimate as the applied strain rate field, $\dot{\varepsilon}_{eq}(x, y, z, t)$, varies significantly with time, t, during the bite (jaw protrusion), and with position (x, y, z) in the food item. In addition, $\dot{\varepsilon}_{eq}(x, y, z, t)$ fundamentally scales up with decreasing food item size and/or with increasing bite speed; the latter varies significantly between breeds and can be influenced by a number of factors (to be discussed in Section 5). Testing at $\dot{\varepsilon}_{eq} > 5/s$ through conventional testing machines was hindered by typical speed limits (here 16.66 mm/s (=1 m/min)). High speed machines (e.g. hydraulic) were not utilised due to typical undesirable noise in the measurements associated with dynamic effects [33], taking also into account the soft nature of the materials investigated here.'

Comment 7: What is the friction law used in the FE model? Is it a simple Coulomb's law? Any critical shear stress before the friction is activated?

Reply: The details of the currently applied law are now provided in Section 4, via the added text:

'The analysis is run parametrically for five friction coefficients: $\mu = 0, 0.15, 0.3, 0.5, 0.8$, associated with the tangential frictional behaviour, while a 'hard' pressure-overclosure relationship is consistently used for the interaction along the normal direction. These conditions are enforced within the classical isotropic Coulomb friction law, assuming that μ is independent of slip rate and contact pressure.'

Note that as we have clarified earlier in correspondence to previous comments, modelling sticky tooth-food interactions through a sophisticated friction law did not lie within the scope of this study. In addition, this would involve significant experimental complexity. This is mentioned in the revised manuscript through the text: 'This highlights that the parameter μ and potentially also the characteristics of the underlined friction law, generally influence significantly the predicted chewing response. However, characterising the details of in-vivo tooth-food interaction, as well as of the friction between the 3D-printed surfaces and the extrudates, can be challenging; this explains why using a simple friction law was instructive for the purposes of this study.'

Comment 8: In Explicit, the simulation results, especially the fracture model, is highly dependent on the mesh size. How is this optimised in the model?

Reply: A similar concern has been raised by the first reviewer and therefore a rigorous mesh sensitivity study is now presented in Appendix-B.

Comment 9: Even though the displacement rate of 1.66 mm/s with $\Delta t = 10$ s are not realistic conditions for mastication (based on the literature values reviewed by the authors), the effectiveness and predictive capability of the proposed FE model can be analysed by plotting the predicted mandible force to compare with the experimental data shown in Figure 4(a). By doing so, the assumption of coefficient of friction of 0.3 can also be verified.

Reply: The authors agree. A similar point has been raised by the 1st reviewer. In the revised manuscript, in Appendix-A we now present additional simulation data obtained both for the lower speed 1.66 mm/s, as well as the high speed of 166.6 mm/s (together with the data previously presented only for 16.6 mm/s); these are obtained for the coefficient of friction 0.3. These additional data indeed justify further the predictive capability of the model at different speeds.

Other comments:

1. There are few typos. The whole manuscript should be rechecked.
 - Page 3, "a large extend" should be "a large extent"
 - Page 9, " $\Delta t_{\{t\}} = 1^{-5}$ s" should be " $\Delta t_{\{t\}} = 10^{-5}$ s"

Reply: The typos have been corrected and the entire manuscript has been rechecked.

2. Page 14, there is no Figure 5(h).

Reply: These errors have been corrected.

2. Page 15, there is no Figure 5(h), this should be Figure 5(g). Also, there is no Figure 13, this should be Figure 6.

Reply: The authors apologise. These errors have been corrected.

REVIEWERS' COMMENTS:

Reviewer #1 (Remarks to the Author):

The authors have addressed all of my comments very well and I have no further comments. I recommend publication of the revised manuscript without further changes.

Congratulations on an exceptional paper!

Reviewer #3 (Remarks to the Author):

The authors have tackled a very challenging research through the use of mechanical testing and computer simulation to understand the first bite (oral processing) and the resulting teeth cleaning efficacy of pet foods. Overall, it is very interesting work and has a direct application in pet food industry as well as a promising potential in pet food formulation design. Although various assumptions were made to simplify the problem, this work will provide a platform to other researchers to further tackle the dynamic process of food oral processing. The authors have responded to all the questions raised very comprehensively. Therefore, I would recommend it for a publication in the Nature Communications.

RESPONSE TO REVIEWERS

Manuscript number: NCOMMS-18-35463-T

'Computer simulations of food oral processing to engineer teeth cleaning'

20/06/2019

Dear editor and reviewers,

We would like to thank you as well as the reviewers for the very positive response to our first revision and for finding our manuscript entitled 'Computer simulations of food oral processing to engineer teeth cleaning', suitable for publication in Nature Communications without further technical changes.

We attach the below feedback provided by the reviewers upon initial submission.

Reviewer 1:

The authors have addressed all of my comments very well and I have no further comments. I recommend publication of the revised manuscript without further changes.

Congratulations on an exceptional paper!

Reviewer 3:

The authors have tackled a very challenging research through the use of mechanical testing and computer simulation to understand the first bite (oral processing) and the resulting teeth cleaning efficacy of pet foods. Overall, it is very interesting work and has a direct application in pet food industry as well as a promising potential in pet food formulation design. Although various assumptions were made to simplify the problem, this work will provide a platform to other researchers to further tackle the dynamic process of food oral processing. The authors have responded to all the questions raised very comprehensively. Therefore, I would recommend it for a publication in the Nature Communications.